# Phytochemicals as Chemo-Preventive and Therapeutic Agents Against Bladder Cancer: A Comprehensive Review

**DOI:** 10.3390/diseases13040103

**Published:** 2025-03-30

**Authors:** Orestis Porfyris, Paraskevi Detopoulou, Theodora Adamantidi, Alexandros Tsoupras, Dimitris Papageorgiou, Anastasios Ioannidis, Andrea Paola Rojas Gil

**Affiliations:** 1Laboratory of Basic Health Sciences, Department of Nursing, Faculty of Health Sciences, University of the Peloponnese, Akadimaikou GK, 3 Building OAED, 22100 Tripoli, Greece; orestisporfyris@yahoo.gr (O.P.); tasobi@uop.gr (A.I.); 2Department of Nutritional Science and Dietetics, Faculty of Health Sciences, University of Peloponnese, New Building, Antikalamos, 24100 Kalamata, Greece; p.detopoulou@uop.gr; 3Hephaestus Laboratory, School of Chemistry, Faculty of Sciences, Democritus University of Thrace, Kavala University Campus, 65404 Kavala, Greece; theadam@chem.duth.gr (T.A.); atsoupras@chem.duth.gr (A.T.); 4Department of Nursing, Faculty of Health Sciences, University of Peloponnese Panarcadian Hospital of Tripoli, Red Cross Terminal (Administrative Services) 2nd Floor, 22100 Tripoli, Greece; dpapageorg@uop.gr

**Keywords:** bladder cancer, signaling pathway, phytochemical, polyphenol, chemoprevention, apoptosis, proliferation

## Abstract

Bladder cancer has a high incidence worldwide and is characterized by a high recurrence rate, metastatic potential, and a significant socioeconomic burden. Conventional treatment modalities usually exhibit serious adverse complications, which also negatively affect patients’ quality of life. In the context of exploring new treatment approaches with fewer side effects, the utilization of natural compounds as alternative and/or complementary therapeutic options seems appealing. In the present study, the potential use and effects of various bioactive phytochemicals, including curcumin, resveratrol, epigallocatechin, genistein, and several others, in bladder cancer treatment are thoroughly reviewed. A special focus is given to their potential to beneficially modulate important molecular signaling pathways and mechanisms affecting cell survival, proliferation, migration, and apoptosis, which play a crucial role in the pathogenesis of bladder cancer, such as the PI3K/AKT/mTOR, Ras/Raf/MEK/MAPK, Wnt/β-Catenin, Notch, Hedgehog, Hippo, JAK2/STAT3, and PAF/PAF-receptor pathways. Nevertheless, most studies have been conducted in cell cultures and animal models. Due to differences in genetics and metabolism, more clinical trials are needed to ensure the bio-efficacy of these phytochemicals in humans.

## 1. Introduction

Bladder cancer (BC) is the ninth most frequent cancer worldwide and is associated with high mortality [1]. The most common type of BC is urothelial cell carcinoma. Although in 75% of cases it presents as a non-muscle-invasive disease (NMIBC) with a good prognosis, it has a high recurrence rate. The remaining 20–30% of cases present as muscle-invasive disease (MIBC), exhibit high metastatic potential, and often have a poor outcome. The appropriate treatment is either aggressive surgical intervention (radical cystectomy) or chemotherapy, which is highly toxic and negatively affects the quality of life [1].

Currently, cancer researchers are focusing on phytochemicals, groups of substances derived from natural products, e.g., polyphenols, which possess both chemopreventive and therapeutic properties against various types of cancer, including BC, to improve prognosis by concurrently supporting conventional therapy. Phytochemicals target specific signaling pathways regarding apoptosis, inhibition of the cell cycle, angiogenesis, proliferation, migration, and adhesion of tumor cells [2]. Several in vitro, in vivo, epidemiological studies, and clinical trials are being conducted to investigate the role of phytochemicals in cancer [3].

Previous works have focused on the role of phytochemicals in BC and their potential mechanisms [4,5] or the role of specific natural substances in cancer in general and in BC specifically [6,7,8,9]. Phytochemicals have a different role in the prevention and treatment of cancer compared to current chemotherapeutic agents. Their use in chemoprevention (defined as the process of preventing, suppressing, or reversing the initial phase of carcinogenesis or preventing the invading potential of premalignant cells) is crucial. Indeed, focusing on carcinogenesis control rather than curing the end-stage disease of cancer is better [10]. The interest in the area of chemoprevention has largely increased with the growing understanding of cancer biology, identification of molecular targets, and the fact that phytochemicals act in different steps of carcinogenesis [11].

Cancer is a multi-stage disease, and most chemotherapeutic drugs target advanced stages. The drawbacks of chemotherapeutic drugs include side effects, high costs, and single-target mechanisms. In terms of single targeting, although single hits are sufficient for certain cancers, in the majority of cases, multi-targeting is needed. Moreover, phytochemicals exhibit a synergistic effect with chemotherapeutic drugs, with a subsequent reduction of cost and adverse events. On the other hand, despite the proven anticancer potential of phytochemicals in epidemiological and preclinical studies, these findings could not be replicated in clinical trials. Some of them were withdrawn due to a lack of evidence and/or risk of harm and indeed well-conducted clinical trials are urgently needed to shed light on the appropriate form and dose to achieve bioavailability and clinical efficacy.

This review aims to provide a fresh overview of the signaling pathways implicated in BC pathogenesis including recent knowledge and to shed light on the most promising phytochemicals against BC. Furthermore, ongoing clinical trials regarding phytochemicals were captured and presented. More specifically, the present review answers the following questions:i.What are the main signaling pathways involved in BC development?ii.What are the main phytochemicals exerting protective actions in BC?iii.What trials are being conducted in humans with BC, regarding the main phytochemicals?

## 2. Signaling Pathways in Bladder Cancer

### 2.1. PI3K/AKT/mTOR

The phosphoinositide 3 kinase/protein kinase B/mammalian target of rapamycin (PI3K/AKT/mTOR) pathway is crucial, as it regulates cell growth, proliferation, and survival, and its de-regulation has been reported in 40% of bladder cancers [12]. The cascade starts with the phosphorylation of PI3K, which leads to the phosphorylation of PIP2 into PIP3. Subsequently, the PIP3 phosphorylates and activates AKT, localizing it to the plasma membrane. AKT partakes in several downstream effects involved in cellular proliferation, leading to the final activation of mTOR complexes 1 and 2 [13]. This pathway is controlled by inhibitory mechanisms (including the phosphatase and tensin homolog (PTEN)). The main mutations observed in BC within this pathway are mutated PTEN or loss of PTEN heterozygosity, and mutation of phosphatidylinositol-4,5-biphosphate 3-kinase catalytic subunit alpha (PIK3CA), mTOR, tuberous sclerosis 1 (TSC1), and AKT genes (Pathway A, Figure 1) [14,15].

Targeting components of this pathway have emerged as a therapeutic strategy. Clinical trials and preclinical studies have evaluated monotherapy or combination therapies using PI3K/AKT/mTOR inhibitors to overcome resistance and achieve better outcomes [16]. Furthermore, genetic variations in this pathway have been associated with survival outcomes in muscle-invasive and metastatic bladder cancer patients. A study analyzing single-nucleotide polymorphisms in this pathway found correlations between specific genetic variations and patient prognosis, highlighting the pathway’s clinical significance [17].

### 2.2. Ras/Raf/MAPK

The Ras (rat sarcoma virus)/Raf (rapidly accelerated fibrosarcoma)/MAPK (mitogen-activated protein kinase) signaling pathway plays a crucial role in transmitting signals from the external environment to the cell nucleus, where it activates specific genes responsible for cell growth, division, and differentiation. This process is regulated by extracellular signal-regulated kinases 1 and 2 (ERK1/2), c-Jun NH2-terminal kinase (JNK), and p38, which function as inhibitory mechanisms. The pathway starts with the binding of an extracellular mitogen (e.g., EGF) to a cell surface receptor. Such an action stimulates Ras activity, followed by kinase Raf’s activation, which in turn phosphorylates and activates MAPK kinase (MEK). MEK activates MAPK, which ultimately triggers multiple transcription factors (e.g., Myc) [18]. Mutations in Ras or the fibroblast growth factor receptor 3 (FGFR3) take place in BC, affecting MAPK activation (Pathway B, Figure 1) [19].

Approximately 20% of urothelial tumors exhibit focal amplification of the *RAF1* gene, leading to the activation of the RAF/MEK/ERK signaling pathway and tumor growth. These tumors are sensitive to RAF inhibitors, suggesting potential therapeutic strategies that have shown efficacy in preclinical studies [20]. Mutations in RAS genes, such as HRAS, occur in a subset of bladder cancers, resulting in constitutive activation of the MAPK pathway and driving uncontrolled cell proliferation. Identifying patients with RAF1 amplifications or RAS mutations can guide personalized treatment strategies and potentially improve clinical outcomes [21].

### 2.3. Hedgehog Pathway

Hedgehog (Hh) is a signaling pathway that transmits information to embryonic cells required for proper cell differentiation. Three ligands, Sonic Hedgehog (SHH), Desert Hedgehog (DHH), and Indian Hedgehog (IHH), are able to bind and inactivate the 12-transmembrane protein Patched (Ptch1). In response to this binding, Ptch no longer inhibits Smoothened (Smo), and this cascade activates the glioma-associated oncogene (Gli), which translocates to the nucleus and triggers target genes [22]. In BC, loss of SHH may coincide with progression to MIBC (Pathway C, Figure 1) [23]. Studies have demonstrated that components of the Hh pathway are overexpressed in bladder cancer tissues, leading to tumor aggressiveness and poorer outcomes [24]. The Hh pathway has also been implicated in promoting Epithelial-to-Mesenchymal Transition in bladder cancer cells, a process that enhances metastatic potential and contributes to drug resistance. Targeting Hh signaling presents a promising therapeutic strategy. Inhibitors of the Hh pathway are being explored for their efficacy in reducing tumor growth and overcoming resistance to conventional therapies [25].

### 2.4. JAK2/STAT3 Pathway

In this signaling pathway, the binding of specific ligands, such as hormones and cytokines, triggers the transphosphorylation of Janus kinase (JAK), which leads to the phosphorylation of tyrosine residues on the associated receptor. This modification creates a docking site for signal transducers and activators of transcription (STATs). Once recruited, JAK phosphorylates STATs, leading to their detachment from the receptor and the formation of homodimers or heterodimers. These then migrate to the nucleus to regulate target gene expression. Research has shown that the JAK2/STAT3 pathway can interact with other signaling networks, including MAPK/ERK and PI3K/AKT/mTOR, to coordinate various cellular functions [26]. Additionally, JAK2/STAT3 activation has been linked to enhanced migration and invasion of breast cancer (BC) cells (Pathway D, Figure 1) [27].

Overexpression of the regulator Musashi-2 has been linked to JAK2/STAT3 activation in 34% of bladder cancer samples. Moreover, upregulation of this pathway, mainly the activation of STAT3, promotes bladder tumor migration and proliferation, as well as direct progression from in situ carcinoma to invasive disease [28]. Inhibitors targeting STAT3/5 have demonstrated antitumor activity in bladder cancer models. Combining STAT3 inhibitors with standard chemotherapeutic agents, such as cisplatin, docetaxel, gemcitabine, and paclitaxel, shows additive effects, suggesting that this combination might be applicable in patients with STAT3-mediated chemoresistance [29].

### 2.5. Wnt/β-Catenin

The Wnt/β-catenin signaling pathway plays a crucial role in controlling cell proliferation, fate determination, differentiation, and polarity. Activation occurs when a Wnt ligand binds to the seven-pass transmembrane Frizzled (Fz) receptor along with its co-receptor, lung resistance protein (LRP). This signaling cascade results in the accumulation of β-catenin in the nucleus, where it triggers the expression of specific genes associated with cell growth and proliferation [30]. Alterations that affect the Wnt/β-catenin pathway in BC are the adenomatous polyposis coli (APC) gene mutation and other alterations of Wnt family molecules [31]. The inactivation of Wnt Inhibitory Factor1 and the loss of other Wnt pathway inhibitors are also significant in BC progression [32]. Moreover, Wnt cross-talks with PI3K and MAPK pathways to promote tumorigenesis (Pathway E, Figure 1) [33]. A three-mRNA signature related to the Wnt pathway (comprising MAPK10, RAC3, and PPP2CB) has been identified as an independent prognostic factor for bladder cancer. This signature is associated with epithelial–mesenchymal transition (EMT) and immune cell infiltration [34] The activation of the Wnt/β-catenin pathway contributes to resistance against immunotherapy and chemotherapeutic agents, such as paclitaxel. Targeting this pathway holds promise for improving treatment outcomes and developing personalized therapeutic strategies [35].

### 2.6. Notch Pathway

The Notch signaling pathway plays a crucial role in cell-to-cell communication. Moreover, it regulates gene expression mechanisms that influence cell differentiation throughout embryonic development and adulthood. This pathway involves one of four transmembrane receptors, NOTCH1-4, and is activated when a ligand binds to a Notch receptor on a neighboring cell. Upon activation, the receptor undergoes cleavage, releasing the intracellular domain of Notch (ICN), which is translocated to the nucleus and alters gene expression. This signaling cascade triggers multiple cellular proliferation pathways, including cyclin D1, c-Myc, p21, survivin, and nuclear factor kappa light chain enhancer of activated B cells (NF-κB) [36]. Its significance in BC is controversial, as it can be both tumor-suppressive and tumor-promoting. Specifically, NOTCH1 may be tumor-suppressive and inhibit the epithelial-to-mesenchymal transition (EMT), while NOTCH2 may be tumor-promoting for the EMT procedure [37]. Moreover, NOTCH3 has been identified as an independent predictor of poor outcomes, suggesting its potential as a prognostic marker (Pathway F, Figure 1) [38]. The dual role of Notch signaling in bladder cancer underscores the importance of assessing the activation status of specific Notch pathway components when considering therapeutic strategies.

### 2.7. Hippo Pathway

The Hippo signaling pathway is essential for regulating cell proliferation, differentiation, and survival, contributing to organ development and homeostasis. This pathway operates as a kinase cascade, where mammalian Ste20-like kinases 1 and 2 (MST1/2) phosphorylate and activate large tumor suppressors 1 and 2 (LATS1/2). The primary function of this cascade is to inhibit the activity of two transcriptional coactivators: Yes-associated protein (YAP) and transcriptional coactivator with a PDZ-binding motif (TAZ), which interact with proteins such as postsynaptic density protein (PSD95), Drosophila disc large tumor suppressor (DlgA), and zonula occludens-1 protein (ZO-1). When YAP and TAZ are active, they move into the nucleus, where they bind to the transcriptional enhancer activator domain (TEAD) family of transcription factors, promoting the expression of genes linked to cell proliferation, survival, and migration [39]. YAP and TAZ play a significant role in breast cancer (BC) progression. Reduced expression of MST1/2 and LATS1 has been observed in localized BC, whereas high levels of YAP and TAZ are associated with high-grade and metastatic BC. Additionally, YAP has been identified as a prognostic marker, indicating poorer outcomes and resistance to cisplatin-based chemotherapy (Pathway G, Figure 2) [40]. Targeting the Hippo-YAP pathway could sensitize cancer cells to chemotherapeutic agents, offering a potential therapeutic strategy. Moreover, YAP1 influences the tumor immune microenvironment by promoting immune evasion. Inhibiting YAP1 may enhance the efficacy of immunotherapies in BC patients [41].

### 2.8. Platelet Activating Factor (PAF)

Platelet-activating factor (PAF) is a potent lipid mediator involved in inflammation and various related pathological situations, including cancer [42]. PAF enhances cell proliferation through β-catenin signaling in hepatocellular carcinoma [43]. Moreover, smoking increases PAF production due to urothelial inflammation and the rise of mutations in nine cancer-related genes [44]. PAF also favors neo-angiogenesis, via the activation of endothelial cells (ECs) and the production of growth factors (GFs), proteases, and cytokines underlying its role in metastasis [45]. Interestingly, the inhibition of PAF or its receptor (PAF-R) has been proposed as a strategy to inhibit bladder tumor cell accumulation [46]. In addition, PAF is involved in various thrombo-inflammatory procedures, such as cardiovascular diseases [47], HIV [48], COVID-19 [49], etc., which have also been linked to BC [50,51]. PAF has been recently reported to be inversely related to cellular health as reflected by phase angle [52], which in turn has been connected to adverse nutritional status in patients with cancer [53]. (Pathway H, Figure 2) [27]. Targeting the PAF-PAF-R interaction could serve as a beneficial therapeutic approach for managing tumor growth and metastasis in bladder cancer patients [46].

### 2.9. Other Signaling Pathways and Important Receptors in Bladder Cancer Pathogenesis

Transforming growth factor-beta (TGF-β) functions as a tumor suppressor in the initial stages of tumorigenesis but later transitions into a tumor promoter as the tumor progresses. Dysregulated TGF-β signaling plays a role in various aspects of cancer development, including tumor initiation, growth, invasion, metastasis, and the remodeling of the tumor microenvironment (TME). It can activate the mTOR pathway in BC, while increased expression of TGF-β receptors is associated with the activation of suppressor of mothers against decapentaplegic (SMAD) proteins, leading to high-grade and muscle-invasive bladder cancer (MIBC) (Pathway I, Figure 2) [54]. Genetic polymorphisms in TGF-β pathway genes may influence bladder cancer susceptibility. A comprehensive case-control study evaluated single-nucleotide polymorphisms (SNPs) in key TGF-β pathway genes, suggesting that these genetic variations could modulate the risk of developing bladder cancer [55]. TGF-β plays a multifaceted role in bladder cancer, influencing tumor behavior, patient prognosis, genetic susceptibility, and therapeutic responses.

Epidermal growth factor receptor (EGFR) and human epidermal growth factor receptor 2 (Her2) are tyrosine kinases, and mutations causing upregulation or amplification are associated with a number of cancers. Both receptors co-exist in localized BC and metastases as well, in more than half of the cases [56], and both correlate with aggressive disease and increased mortality [57]. Increases in EGFR expression and subcellular localization have been linked to resistance against cisplatin-based chemotherapy in bladder cancer. This indicates that EGFR may play a role in the development of chemoresistance, impacting treatment efficacy [58]. Increased lymph node invasion, tumor stage, and poor disease-specific survival are observed in BC with overexpression of Her2/neu [59]. However, traditional anti-HER2 therapies have not shown significant clinical benefits in advanced urothelial carcinoma, indicating the need for novel therapeutic strategies [60].

Fibroblast growth factor receptor (FGFR) is a tyrosine kinase receptor implicated in cell growth, cell migration, cell survival, and angiogenesis. FGFR1 and FGFR3 are the most frequently mutated isotypes in BC, with FGFR3 mutations occurring in 50–60% of NMIBC and 10–15% of MIBC cases [61]. FGFR mutation indicates a better prognosis, with longer survival and a lower risk of recurrence and progression in BC (Pathway J, Figure 2) [62]. Erdafitinib, a broad-spectrum FGFR inhibitor, has been approved for the treatment of metastatic urothelial carcinoma with FGFR3 alterations, representing a major advancement in targeted therapies for bladder cancer [63]. New FGFR3-selective inhibitors, such as TYRA-300, have shown encouraging early-phase clinical trial results, suggesting potential for more effective and safer treatment options [64].

Vascular endothelial growth factor (VEGF) and its receptor expression promote tumor angiogenesis in BC and are associated with aggressive disease, vascular invasion, and worse prognosis (Pathway K, Figure 2) [65]. Several therapies targeting VEGF or its receptors have been explored. However, VEGF inhibitors have shown mixed results in clinical trials, emphasizing the need for better patient selection, combination strategies, and improved biomarker-driven approaches to optimize treatment outcomes [66].

NF-κB on the other hand, is a transcription factor activated mainly during inflammation and hypoxia. In BC, chemical carcinogens (such as cigarette smoke) induce persistent NF-κB and predispose it to carcinogenic aberrations [67]. NF-κB binds to the inhibitor of NF-κΒ (IκΒ) in the cytoplasm, promotes the expression of inflammatory mediators linked to BC, such as interleukins 8, 5, and 20 (IL-8, IL-5, and IL-20, respectively) [67], and increases the transcription of metalloproteinases 2 and 9 (MMP2 and MMP9, respectively), which are responsible for promoting invasion and metastasis of bladder tumors [68]. NF-κΒ also increases the manifestation of anti-apoptotic genes, including survivin, the cellular inhibitor of apoptosis protein 1 and 2 (cIAP-1/2), and the linked inhibitor of apoptosis protein (XIAP) [69], and is implicated in the overexpression of cyclooxygenase 2 (COX-2) in normal and tumor cells. The overexpression of COX-2 is, in turn, associated with increased invasion, recurrence, and poorer prognosis (Pathway L, Figure 2) [70]. NF-κB inhibitors, such as the dehydroxymethyl derivative of epoxyquinomicin (DHMEQ), have been shown to induce tumor growth suppression and overcome cisplatin resistance in bladder cancer models [71].

Some additional pathways have been identified in studies in which phytochemicals effectively inhibit cancer signaling pathways and prevent cancer progression. The first relates to the restoration of microRNA (miRNA) function. MicroRNAs (miRNAs) are short, non-coding RNAs, ranging from 18 to 24 nucleotides, that can downregulate target genes at the post-transcriptional level. They play a crucial role in various biological processes, including apoptosis, differentiation, and proliferation, with their dysregulation often observed in cancer [72]. The second pathway involves targeting cancer stem cells (CSCs), which are a distinct subpopulation of cells within tumors. CSCs are defined by their ability to self-renew and differentiate into non-tumorigenic progeny, while still contributing to tumor growth. They are known for their resistance to standard anticancer treatments and their involvement in tumor recurrence or relapse (Figure 2) [73].

The abnormal expression profiles of non-coding RNA in BC may be a potential biomarker for early BC detection. Certain microRNAs are associated with tumor aggressiveness and recurrence ratios while the expression of certain long non-coding RNAs has been linked to survival rates. These non-coding RNAs aid in risk stratification, diagnosis, and disease trajectory [74].

## 3. Phytochemicals Targeting Bladder Cancer

### 3.1. Curcumin

Curcumin is a cinnamate derivative isolated from turmeric with known antioxidant, anti-inflammatory, and antitumor properties [75]. It is found in the rhizomes of several Zingiberaceae and Araceae plants [76]. Despite the promising results, its therapeutic application is limited, due to poor absorption, and aqueous solubility and fast metabolism and elimination. The serum concentration of crude curcumin varied significantly, ranging from 1 to 3200 ng/mL, depending on the administered dose (2 to 10 g) and individual physiological differences. In certain instances, even with a 3.6 g dose, curcumin levels fell below the detectable limit (<1 ng/mL). Therefore, utilizing high-performance analytical techniques is essential for accurately determining curcumin’s pharmacokinetic profile [77]. Certain pharmaceutical technologies or unique combinations with other compounds, like piperine or lecithin, have been shown to increase its solubility, prolong its presence in plasma, and enhance both its pharmacokinetic profile and cellular absorption. Some strategies intend to improve its bioavailability through cyclodextrin inclusion complexes, solid dispersions, and nanocarriers with modest success [78].

Various innovative delivery systems, including solid lipid particles, micellar formulations, and hydrophilic nanoparticles, have the potential to boost curcumin concentration by 15 to 20 times. The determination of curcumin pharmacokinetic profile seems to be challenging, due to the low circulant concentrations.

Curcumin targets multiple signaling pathways in bladder cancer. It inhibits, in BC cells (BCCs), human trophoblastic cell surface antigen 2 (Trop2) signaling. Trop2 is a transmembrane glycoprotein that acts as an intracellular calcium signal transducer, and it is overexpressed in BC [79]. Curcumin was observed to reduce BCC motility and proliferation, and it also inhibits p27 and cyclin E1, inducing G2/M cell cycle arrest [80]. Curcumin reduces the expression of the Wnt/β-catenin signaling pathway, leading to the suppression of EMT [81]. Curcumin administration also downregulates MAPK/ERK5/activator protein 1 (AP-1) signaling, leading to inhibition of EMT and decreased migration and malignancy of BCCs [82]. Curcumin stimulates apoptosis and suppresses cancer stem cells by inhibiting the expression of the Sonic Hedgehog signaling pathway [83]. In combination with melatonin, curcumin downregulates NF-κB and COX-2, which in turn inhibits the proliferation and viability of cancer cells and induces apoptotic cell death [84]. It also reduces the metastatic potential of cancer cells by downregulating MMP2 and MMP9 and upregulating the Tissue Inhibitor of Metalloproteinase-2 (TIMP-2) [85]. Curcumin potentiates the antitumor effect of the Bacillus Calmette–Guerin (BCG) in bladder cancer (by downregulating NF-κB and upregulating tumor necrosis factor (TNF)-related apoptosis-inducing ligand (TRAIL) receptors (TRAILRs)) [86]. It modulates several microRNAs, such as miR-1246, miR-7641, and miR-203, resulting in diminished proliferation and enhanced apoptotic cell death [87,88]. It targets the oncoprotein Aurora A, resulting in reduced proliferation and invasion of bladder cancer cells [89]. The antitumor effect of curcumin is also manifested through targeting the insulin-like growth factor 2 receptor (IGF2) expression and the resultant PI3K/AKT/mTOR cascade [90], the upregulation of the pro-apoptotic factor B-cell lymphoma protein 2 (Bcl-2)-associated X (BAX), and the downregulation of the anti-apoptotic factors Bcl-2 and survivin [91]. Furthermore, curcumin downregulates YAP and TAZ (two key factors of the Hippo pathway), which leads to reduced malignancy and proliferation of BCCs [92]. Recent data suggest that curcumin inhibits the proliferation and migration of BCCs, as well as the AKT/MMP14 pathway [93]. Finally, curcumin enhances the antiproliferative impact of chemotherapeutic agents (5-fluorouracil, paclitaxel) [94,95] and, combined with cisplatin, targets multi-drug resistance (MDR) pathways by causing a decrease in nuclear Nrf2 levels (Nuclear factor erythroid 2-related factor 2) [96] and an increase in KEAP1 levels (Kelch-like ECH-associated protein 1). This modulation suggests that curcumin downregulates the Keap1-Nrf2 pathway, which is known to play a role in cellular defense mechanisms against oxidative, electrophilic stress and chemotherapeutic agents. Under normal conditions, KEAPT1 functions as part of an E3 ubiquitin ligase complex, strictly controlling NRF2 activity through ubiquitination and degradation via the proteasome. Under stress conditions, KEAP1 prevents NRF2 degradation. Once stabilized, NRF2 can activate the transcription of antioxidant genes [97] (Figure 3). Curcumin has also been found to enhance radiosensitivity by increasing DNA damage and inhibiting repair mechanisms in cancer cells, but also protects normal cells from radiation-induced damage [98]. In parallel, potential interference with cancer therapy was also observed. Since curcumin is a potential antioxidant, it may neutralize ROS and thus potentially reduce the effect of some treatments that are based on ROS to kill cancer cells. However, its antioxidant effects and drug interactions require careful evaluation. One of the key advantages of curcumin for therapeutic applications is its low toxicity, as doses of up to 10 g per day have been shown to cause no adverse effects [99].

### 3.2. Epigallocatechin-3-Gallate (EGCG)

EGCG is a polyphenolic compound found in the leaves of green tea and most health benefits of tea are attributed to it, since it is believed to possess antioxidant, anti-inflammatory, antibacterial, and anticancer properties. EGCG has relatively low oral bioavailability. After consumption, its plasma concentrations are detectable but present variability among individuals, with mean peak plasma concentrations typically occurring within 1.7 h. The plasma half-life of EGCG is approximately 5 h. Minimal amounts of EGCG are excreted in urine, showing efficient transport to the liver or metabolism by gut microbiota into other bioactive compounds present in plasma, bile, and urine. Εxtensive research has shown that pharmacokinetic parameters of EGCG exhibit considerable variability across studies and individuals. Several factors, including storage conditions, fasting, albumin levels, vitamin C, fish oil, and piperine, have been shown to enhance plasma concentrations and overall bioavailability [100]. In contrast, its bioavailability is reduced by oxidation, sulfation, glucuronidation, gastrointestinal degradation, and interactions with Ca^2+^, Mg^2+^, and trace metals, as well as genetic factors like COMT polymorphisms [101]. In order to improve EGCG bioavailability, various strategies have been used including its coadministration with piperine, which increases EGCG plasma concentration through the inhibition of its metabolic degradation. Nanoparticles, liposomes, and micelles can also enhance EGCG stability and its absorption in the gastrointestinal tract [102].

EGCG inhibits the proliferation and promotes apoptosis of BCCs by activating caspases 8,9, and 3, Bax, Bcl-2, and Rickettsia ankyrin repeat protein (RARP) and downregulating NF-κB and MMPs [103]. Another mechanism through which EGCG acts is via a reduced expression of the PI3K/AKT pathway [104,105] and suppression of ERK1/2, JNK, and AP-1 [105]. In addition, it inhibits growth and induces apoptosis of BCCs by restoring tissue factor pathway inhibitor 2 (TFPI-2) expression [106]. The anti-inflammatory properties of EGCG, manifest through the targeting of the toll-like receptor 4 (TLR4) signaling pathway and the intervention at the genes with potential miRNA interactions, particularly tensin 1 (TNS1), alpha-1,6-mannosylglycoprotein 6-beta-N-acetylglucosaminyltransferase B (MGAT5B), and beta-arrestin 1 (ARRB1) [107]. EGCG inhibits bladder cancer stem cells (BCSCs) via suppression of the sonic hedgehog pathway, as demonstrated by the reduced expression of bladder CSC markers (CD44, CD133, Oct4, ALDH1, and Nanog). Finally, doxorubicin exhibits improved efficacy in addition to EGCG in BC via the NF-κΒ/murine double minute 2 (MDM2)/p53 pathway [108]. Moreover, the immune response of cytotoxic lymphocytes and dendritic cells (DCs) is augmented by EGCG, while the immunosuppression of myeloid-derived suppressor cells (MDSCs) and regulatory T cells (T-regs), is reduced [109]. Additionally, EGCG may notably affect glucose uptake, carbohydrate and lipid metabolism during BC (Figure 3) [109].

Toxicological and human safety data from tea preparations consumed in solid bolus form were used to determine a safe intake level of 338 mg of EGCG per day for adults. Based on reported adverse event data in humans, a proposed observed safe level (OSL) of 704 mg of EGCG per day could be established for tea preparations in liquid form. Incorporating EGCG into cancer treatment regimens, either as a standalone agent or in combination with existing therapies, holds promise. However, comprehensive clinical studies are essential to fully elucidate its efficacy and safety profiles in patients with BC cancer [110].

### 3.3. Genistein

Genistein is a soy isoflavone, like glycitein and daidzein, and it is classified as a phytoestrogen. Genistein has low solubility, and it may not be well absorbed in high doses without proper formulation. Extensive first-pass metabolism is the major reason for low oral bioavailability, together with the high expression levels of efflux transporters, particularly BCRP [111]. Several approaches have been shown to improve genistein solubility and bioavailability. Some examples include the use of cocrystals of genistein with piperazine, encapsulation of genistein in solid lipid nanoparticles, and the use of excipients [112].

Genistein is able to inhibit cancer cell growth, survival, metastasis, and angiogenesis by inducing apoptosis via the damage of the DNA of the cancer cell [9]. In particular, genistein was found to cause a dose-related inhibition of BC proliferation, induction of apoptosis, and G2/M phase arrest, in synergy with hydroxycamptothecin (HCPT), a drug that inactivates DNA topoisomerase I [113]. A phase II clinical trial in pre-surgical patients with BC showed that genistein decreased EGFR phosphorylation and downstream phosphorylated MAPK in cancer cells, although this was not translated in notable differences as far as tumor proliferation, apoptosis, and apoptotic-inhibiting markers are concerned [114]. Further clinical trials could contribute to the application of soy phytochemicals as potent prevention regimens for BC progression (Figure 3) [115].

Genistein shows promise as an adjunct in bladder cancer treatment, particularly in reducing side effects associated with conventional therapies and targeting specific cancer cell pathways. However, its potential interactions with other treatments and possible toxicities necessitate further clinical evaluations to ensure its safe and effective integration into cancer therapy approaches [116].

### 3.4. Resveratrol

Resveratrol (RSV) is a non-flavonoid polyphenolic compound found naturally in grapes, berries, peanuts (especially Polygonum cuspidatum), and other foods, such as soybeans. Among its numerous biological activities, it has antitumor properties [117]. Even though approximately 70% of orally ingested resveratrol is absorbed in the intestine, its bioavailability is low, around 0.5%, due to its extensive first-pass metabolism in the liver and intestine, where RSV undergoes fast glucuronidation and sulfation, leading to the formation of metabolites with limited biological activity. Various strategies have been explored to improve its systemic exposure, such as its encapsulation in nanoparticles [118], structural chemical modifications to increase its stability and resistance to metabolic processes, and co-administration with other components to reduce its rapid conjugation [119].

Studies have demonstrated that RSV has multiple effects on BCCs, as, at first, it can cause G1 cell cycle arrest through p21 and p38 activation [120]. RSV may induce the disruption of the mitochondrial membrane via excessive ROS production, which leads to the activation of caspase 3 and 9 and ends in cell death [121]. Reactive oxygen species (ROS) production is also associated with DNA fragmentation and subsequently reduced cancer cell proliferation [122]. A key therapeutic target of RSV-induced apoptosis is miR-21. This microRNA, which is often overexpressed in various cancers, plays a crucial role in promoting cell survival and proliferation while inhibiting apoptosis. The inhibition of miR-21 leads to a downstream reduction in AKT phosphorylation, a process critical for activating the PI3K/AKT signaling pathway, inducing cell apoptosis [123]. The same result is achieved as RSV interferes with inhibiting the STAT3 signaling pathway, which regulates a variety of tumor-promoting gene promoters, such as VEGF, c-MYC, cyclin D1, and survivin, and induces the S phase cell cycle arrest by activating the silent transformation regulator 1-tumor protein p53 (Sirt1-p53) pathway involved in the nuclear translocation of Sirt1 and p53. The final result is reduced cell growth, apoptosis, and S-phase cell cycle arrest in BC [124]. RSV additionally decreases the phosphorylation of JNK1/2 and ERK1/2, thus inhibiting the expression of MMP2 and MMP9 and suppressing cell migration and invasion [125]. Wang et al. [126] showed that RSV can reverse drug resistance by decreasing multidrug resistance protein 1 (MRP1), LRP and glutathione S-transferase (GST), and increasing topoisomerase-II (Topo-II) expression levels [126]. RSV treatment can decrease the expression of the polo-like kinase 1 (PLK1) gene, which is associated with cell cycle arrest at the S phase [127], and modulates the expression of Ras association domain family protein 1 isoform A (RASSF1A) and homeobox B3 (HOXB3) genes, leading to reduced proliferation of BCCs [128]. Almeida demonstrated that the RSV-induced apoptosis is associated with reduced expression of the SRC (sarcoma) gene and reduction of the AKT/mTOR pathway [129]. Recently, the antitumor effect of RSV was demonstrated, via ROS production, cell cycle modulation, and inhibition of BC migration [130]. A biphasic dose–response phenomenon has also been suggested for RSV, since, in high doses, it displays an antiproliferative effect, and in low doses, it has an anti-angiogenic effect [131].

Finally, RSV was studied in conjunction with cytotoxic drugs and other compounds in BC. In this context, Alayev [132] showed that RSV, combined with rapamycin prevented the rapamycin-induced reactivation of AKT, causing apoptosis, decreased cell migration, and clonogenic survival [132]. Cho showed that combining RSV with gemcitabine could overcome gemcitabine resistance in BC [133]. In addition, Soares et al. [134] demonstrated that when RSV was co-administered on BCCs with a low concentration of doxorubicin, the antitumor and anti-proliferative effects of the drug were enhanced (Figure 4) [134].

According to clinical studies, resveratrol is generally considered safe and well tolerated as a single dose up to 450 mg/day for a 60 kg person without adverse effects. However, 1000 mg daily or higher doses were reported to inhibit cytochrome P450 enzymes, leading to interactions with many drugs. Higher doses were shown to elevate some biomarkers of CVD risk (ox-LDL, sE-Selectin 1, sICAM-1, sVCAM, tPAI-1) and affect the gastrointestinal system. These adverse effects could vary according to individual gut microbiota, health status, gender, lifestyle, and the form of administration (with or without food, caplet, tablet, powder, etc.) [135].

### 3.5. Sulforaphane-Erucin

Sulforaphane and erucin (sulfone analogue of sulforaphane) are isothiocyanates derived from cruciferous vegetables like broccoli, Brussels sprouts, cabbage, cauliflower, etc. Their bioavailability depends on myrosinase-mediated conversion, and it is highly influenced by food preparation and gut microbiota. Even though they are efficiently absorbed, extensive first-pass metabolism in the liver leads to the formation of several conjugated metabolites with a longer half-life contributing to their overall biological activity. However, detailed pharmacokinetic parameters, such as their precise half-life, and clearance rates, require further investigation [136].

An epidemiological study confirmed a dose-related reduction in BC risk with the consumption of cruciferous vegetables, whose antitumor properties are attributed to sulforaphane and erucin [137]. Sulforaphane notably induces GST, nicotinamide adenine dinucleotide phosphate (NAD(P)H), and quinone oxidoreductase, which are important protective enzymes against oxidants and carcinogens, and therefore inhibit bladder carcinogenesis [138]. It induces apoptosis and mitotic arrest either via mitochondrial disruption, cleavage of caspase 3,9, poly(ADP-ribose) polymerase (PARP), and cytoplasmic aggregation of histone-associated DNA fragments [139], or via ROS-associated processes [140]. Another mechanism of apoptosis, mediated by dysregulation of the function of mitochondria, is through the release of cytochrome C and Bcl-2-related pathways [141]. Sulforaphane can epigenetically modulate the expression of genes involved in apoptosis and proliferation. This is accomplished by inhibiting histone deacetylase (HDAC) activity and DNA methyltransferase (DNMT) expression [142]. Sulforaphane downregulates survivin, EGFR, and Her2/neu and leads to G2/M cell cycle arrest and apoptosis [143]. It can modulate angiogenesis either via the inhibition of VEGF, transcription factor hypoxia-inducible factor-1a (HIF-1a), and c-Myc [144], or via the blockage of NF-kB and the reduced expression of COX-2, which is associated with the suppressed transcription of MMP and decreased EMT [145]. Sulforaphane protects the normal bladder cell against chemical-induced DNA damage via activation of Nrf2, which in turn activates cytoprotective genes [146]. Moreover, it has an inhibitory effect on the adhesion and migration of drug-resistant BC cells [147], on the proliferation of drug-sensitive and resistant BC cells [148], and on the formation of pseudopodia [149]. As recently proposed, sulforaphane also inhibits hexokinase 2, an enzyme participating in glycolysis, while it reduces oxidative phosphorylation in mitochondria [150]. Finally, the combination of acetazolamide with sulforaphane is superior in inhibiting, with great efficacy, the PI3K/AKT/mTOR pathway compared to its sole administration [151]. Similar findings regarding the inhibition of BC proliferation and induction of apoptosis have been shown both in in vivo and in vitro studies (Figure 4) [143]. Resistance to TNF-related apoptosis-inducing ligand (TRAIL) is a common challenge in bladder cancer that hinders apoptosis. Notably, sulforaphane treatment counteracts this resistance in bladder cancer cells by increasing ROS production, enhancing levels of cleaved PARP, cleaved Bid, and death receptor 5 (DR5), which in turn leads to a loss of mitochondrial membrane potential (ΔΨm) and triggers apoptosis. These findings indicate that combining sulforaphane with TRAIL could serve as an effective adjunct chemotherapeutic approach for patients with TRAIL-resistant bladder carcinoma. Overall, the studies underscore sulforaphane’s potential as a chemosensitizer and adjunct therapy, offering a promising strategy for managing bladder cancer [152].

Both sulforaphane and erucin demonstrate low toxicity when consumed at doses obtained from a regular diet or standard supplementation. Their rapid metabolism and excretion further support a favorable safety profile.

### 3.6. Extra Virgin Olive Oil (EVOO) Phenols

The phenolic fraction of EVOO (0.4–5% of the olive fruit) consists of more than 30 distinct compounds belonging to different classes, e.g., secoiridoids, such as oleuropein and ligstroside, phenolic alcohols such as hydroxytyrosol and tyrosol, flavonoids, and lignans. Extra virgin olive oil polyphenols undergo extensive metabolic processing in the body. They primarily circulate and are excreted in the urine as conjugated compounds—such as glucuronides, sulfates, and methylated derivatives—while the aglycone forms exhibit relatively poor bioavailability compared to these metabolites [153]. Despite this, olive oil phenols are proven to have antitumor properties against various types of cancer, both in in vivo [154] and in vitro studies [155]. In a study by our group in patients with chronic lymphocytic leukemia, 20 mL of olive oil rich in oleocanthal and oleacein increased caspase-cleaved cytokeratin-18 (ccK18) and apoptosis antigen 1 (Apo1)-Fas, were implicated in apoptosis and decreased survivin and cyclin D [156].

As far as BC is concerned, an epidemiological study showed that regular consumption of olive oil has a protective effect against BC [157]. Coccia et al. [158] showed that EVOO can affect the viability of BC cells in a dose-dependent manner and further inhibit their migration and invasion by modulating the activity of MMP2 [158]. In a later study, the same author demonstrated that EVOO blocks cell cycle progression at the G2/M phase and induces apoptosis, decreases intracellular ROS production, and positively affects paclitaxel cytotoxicity [159]. Similarly, a polyphenol-rich olive oil by-product (mill waste), had an anti-proliferative effect on both chemo-sensitive and resistant BCCs [160]. In another in vitro study with several different BC cell lines, EVOO triggered a form of autophagy and induced cancer cell apoptosis via activating caspases 3 and 9 (Figure 4) [153].

### 3.7. Other Phytochemicals

A variety of compounds have been studied for their chemo-preventive and therapeutic action against BC. Genipin, derived from the cape jasmine plant, was found to inhibit BC augmentation in vivo and induce apoptosis by hindering the PI3K/AKT pathway and activating Bax and cytochrome C. Parthenolide, a sesquiterpene lactone, inhibits proliferation and leads to apoptosis and G1-phase arrest in BC cell lines, by PARP induction and downregulation of Bcl-2 [161]. Moreover, in animal models, dimethylaminoparthenolide improved cisplatin efficacy [162]. Dioscin, used in traditional Chinese medicine, inhibits proliferation and induces apoptosis by reducing gene methylation of death-associated protein kinase 1 (DAPK-1) and RASSF-1a, which are mediators involved in programmed cell death [163]. Kaempferol, a flavonoid found in ginger, reduces phosphorylated AKT and downregulates cyclin D1, suppresses proliferation, and induces apoptosis [164]. Pterostilbene, found in blueberries, grapes, and cranberries, suppresses proliferation and promotes autophagy by hindering the AKT/mTOR/p70 ribosomal S6 kinase (p70S6K) pathway and inducing the MEK/ERK1/2 pathway [165]. Baicalein, a flavone derived from the herb Huang Qin, exerts its effect in BC cell lines via decreased expression of anti-apoptotic factors, increased ROS production, and activation of caspases [166,167]. Quercetin, (along with its analog Isoquercetin) derived from the American cranberry, inhibits proliferation, induces apoptosis, and reduces migration of cell lines through increased phosphorylation of adenosine monophosphate-activated protein kinase (AMPK) [168]. Moreover, it inhibits cell growth via the AMPK/mTOR pathway and induces DNA damage [169]. Vitamin E includes compounds belonging to tocopherols and tocotrienols, which have been studied both in vivo and in vitro studies. In vivo, alpha- and gamma-tocopherol impeded tumor growth through reduced NF-κΒ translocation, and its co-administration with paclitaxel produced a greater effect than either compound alone [170]. In vitro, delta-tocotrienol prevents proliferation and promotes apoptosis by suppressing STAT3 phosphorylation [171]. However, a clinical trial showed a rather harmful effect of vitamin E in patients suffering from BC (Figure 5) [172].

## 4. Ongoing Trials with the Use of Phytochemicals

The clinicaltrials.gov database was searched to identify ongoing trials in patients with BC. More specifically, each phytochemical was entered in the search entry “intervention”, while the key terms “bladder cancer” and “bladder carcinoma” were entered in the search entry “condition/disease”. For phytochemicals, the following terms were used: “curcumin”, “curcuminoid”, “epigallocatechin-3-gallate”, and “EGCG”. “Green tea”, “sulforaphane”, “broccoli”, “erucin”, “extra-virgin olive oil”, “olive oil”, “oleocanthal”, “tyrosol”, “genipin”, “parthenolide”, “dioscin”, “kaempferol”, “pterostilbene”, “baicalein”, “quercetin”, “vitamin E”, and “tocopherol”. The results regarding ongoing clinical trials are summarized in Table 1. For trials completed for more than 3 years or trials withdrawn due to budget constraints (NCT03517995), no data are reported. Only two clinical trials were identified: for curcumin and genistein.

## 5. Discussion

The promising data point to the role of phytochemicals in modulating several molecular pathways implicated in BC. The identification of molecular targets is of utmost importance in the era of personalized medicine, along with genomic and proteomic analysis. In the present review, several molecular pathways are presented, such as PI3K/AKT/mTOR, Ras/Raf/MEK/MAPK, Wnt/β-catenin, Notch, Hedgehog, Hippo, JAK2/STAT3, PAF, and many others. Of course, each substance may have more than one underlying mechanism through which it may modulate cancer risk, and this feature may enhance its “effectiveness”.

In addition, phytochemicals constitute a good candidate for new drug formation or can be used as supplemental therapy. Indeed, phytochemicals can be extracted from their natural origin or chemically synthesized. For example, paclitaxel, derived from yew trees, has already been approved by the Food and Drug Administration (FDA) for cancer therapy [194] and has entered clinical trial stage III for BC in co-administration with other drugs [195]. In addition, most phytonutrients are metabolized and excreted via urine and, thus, are present in the bladder tissue [4].

The urinary microbiome may also modulate response to therapy and be linked to cancer initiation and prognosis [196]. Indeed, the urinary microbiome modulates the bladder microenvironment and influences immunity [197], inflammatory status, and genotoxicity [198], while probiotics may reduce the probability of cancer recurrence after surgery [199]. Phytochemicals, dietary patterns, and lifestyle may modulate urinary microbiome cancer risk [198] and further enhance human health. In turn, each subject’s genetic makeup and microbiome influence the metabolic fate of phytochemicals [200].

Epidemiological studies have shown that patterns rich in meat, saturated fats, eggs, and coffee may be positively associated with BC, while foods rich in phytochemicals, such as cruciferous vegetables, fruits, and green tea, are inversely related to BC risk [201,202]. In fact, a diet rich in antioxidants targets molecular cancer-involved cascades [203,204], and other biomarkers affecting the inflammatory milieu, like glucose [205].

Patients with cancer experience several changes during therapy, in which diet may also play a beneficial role. For example, body composition and muscle mass may deteriorate during therapy [206] and antioxidants [53], or holistic dietary patterns may help protect against sarcopenia [207]. However, few studies have investigated the effects of lifestyle variables on sarcopenia in patients with BC [208].

Because BC is one of the most common malignancies and remains one of the most lethal, and few novel therapeutic modalities have been discovered lately for the treatment of BC (e.g., programmed cell death protein 1 (PD-1) inhibitors), it seems very attractive to use the anticancer properties demonstrated at the molecular level and the remarkable low toxicity of natural compounds as additional chemo-preventive and therapeutic options for BC patients.

Several limitations should be considered in the study of anticancer phytochemicals. The dose, bioavailability, stability, potential side effects, and/or resistance need further research to be determined; for example, RSV has differentiated actions in low and high doses [131]. Another challenge in using phytochemicals must be addressed concerning the variability in natural sources and extraction methods, which could lead to inconsistencies in the concentration and composition of phytochemical preparations. Standardization must be addressed to establish effective and safe dosing regimens. More detailed investigation is also required to explore the mechanisms and strategies for enhancing the accessibility and bioavailability of phytochemicals.

Multidrug resistance (MDR) is a major obstacle in the treatment of human cancer with chemotherapy and it is attributed to ABC transporter proteins, which behave as an energy-dependent efflux pump of anticancer agents. Natural phytochemicals could modulate the expression of these proteins and reverse the chemoresistance of cancer cells. Resveratrol inhibits efflux transporters such as P-glycoprotein (P-gp), multi-drug resistance-associated protein (MRP), and breast cancer resistance protein (BCRP) and is a potential multi-drug-resistant regulator [209]. Similarly, EGCG modulated the function of P-glycoprotein and increased the intracellular accumulation of the chemotherapeutic agent doxorubicin (DOX) in drug-resistant KB-A1 cells [210]. Paradoxically, a different action was observed for genistein. Rigalli et al. showed in their study that genistein increased protein levels of P-glycoprotein and MRP2 in hepatocellular cancer cells in a concentration-dependent manner, and patients with HCC under chemotherapy should abstain from soy-rich diets or dietary supplements, in order to avoid enhanced chemoresistance [211].

Of note, interactions between plant-based or classic drug therapies and phytochemicals cannot be excluded [12], with the example of curcumin and chemotherapeutic agents (5-fluorouracil, paclitaxel) [94,95]. These interactions need to be thoroughly studied both experimentally and clinically to optimize efficacy, reduce drug resistance, and lower the risk of relapse. The timing and duration of phytochemical administration and the effects of their synergistic combinations also need extensive standardization, considering the varying effects observed in trials with both short-term and long-term treatment regimens.

Last but not least, most of the studies were performed in cell cultures and animal models, and the encouraging results have to be confirmed in clinical trials, so as to ensure that these results will hold for humans, due to differences in genetic and metabolic profiles. However, only a few clinical trials are being conducted at the moment, unfortunately.

Simulation analysis can help identify further mechanisms of phytochemicals, as was recently conducted with curcumin and BC [212]. In the future, emerging data from basic research in conjunction with clinical and epidemiological data may be embraced in a more sophisticated way through artificial intelligence (AI) algorithms, which could provide personalized, custom-based solutions in patient treatment [213,214].

Although preclinical data are promising, there is a need for well-designed clinical trials to establish and confirm the safety and efficacy of phytochemicals in bladder cancer patients according to individual characteristics such as genetic background, microbiota composition, etc. The evaluation of long-term effects, potential toxicity at therapeutic doses, and optimal combination strategies with existing treatments is pivotal.

## Figures and Tables

**Figure 1 diseases-13-00103-f001:**
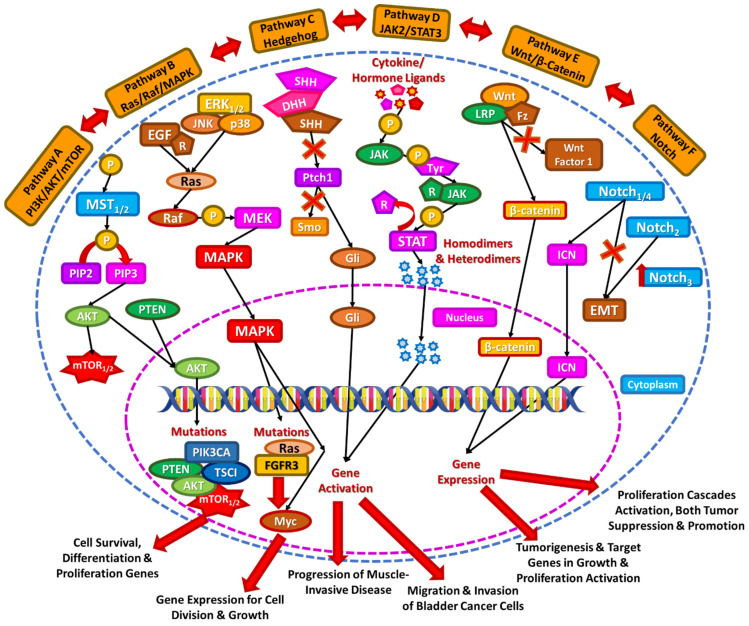
Signaling pathways in bladder cancer, including PI3K/AKT/mTOR, Ras/Raf/MAPK, Hedgehog, JAK2/STAT3, Wnt/β-catenin, and Notch signaling cascades.

**Figure 2 diseases-13-00103-f002:**
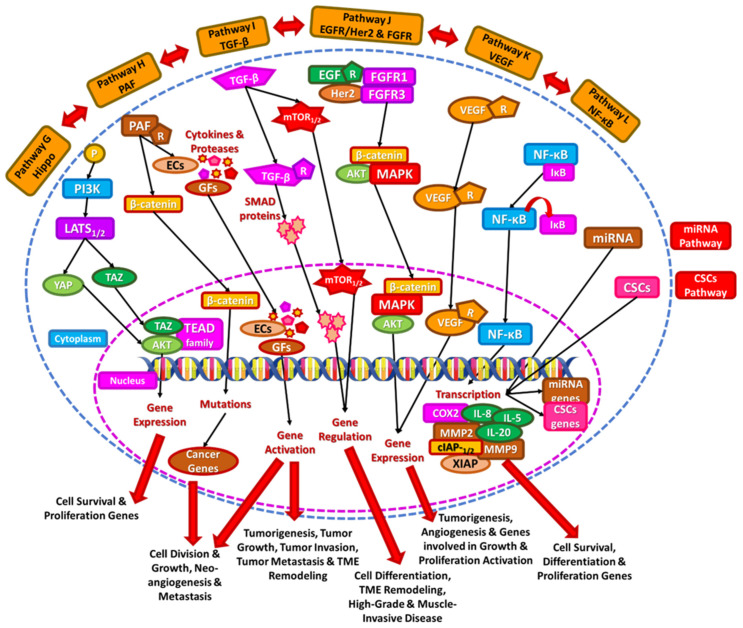
Signaling pathways in bladder cancer including Hippo, PAF, TGF-β, EGFR/Her2 and FGFR, VEGF, NF-κΒ, miRNA, and CSCs signaling cascades.

**Figure 3 diseases-13-00103-f003:**
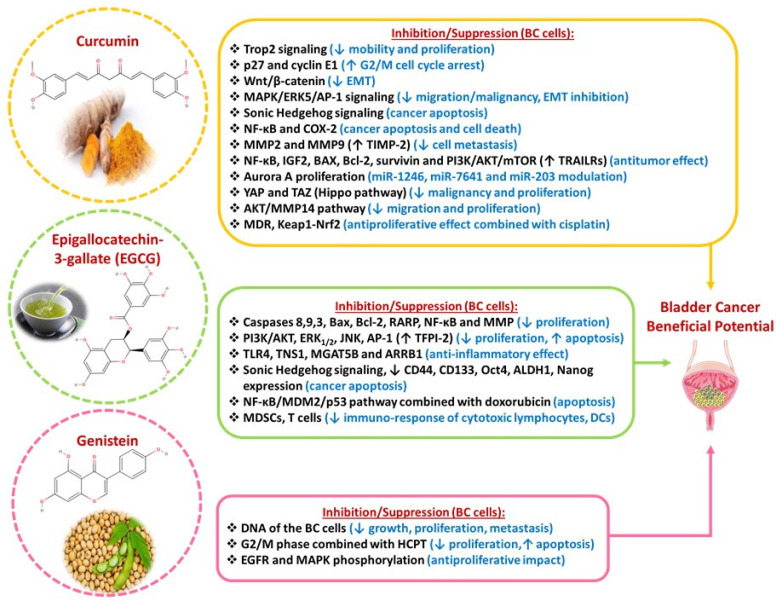
Bladder cancer beneficial potential of curcumin, EGCG, and genistein phytochemicals.

**Figure 4 diseases-13-00103-f004:**
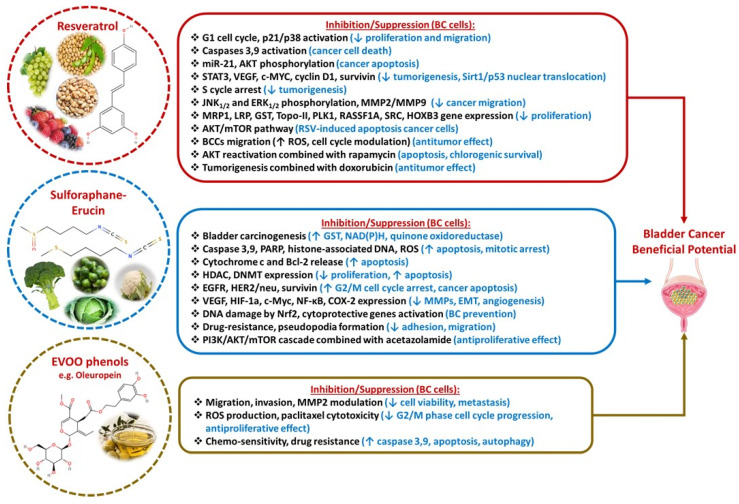
Bladder cancer beneficial potential of Resveratrol, Sulforaphane-Erucin, and EVOO phenol phytochemicals.

**Figure 5 diseases-13-00103-f005:**
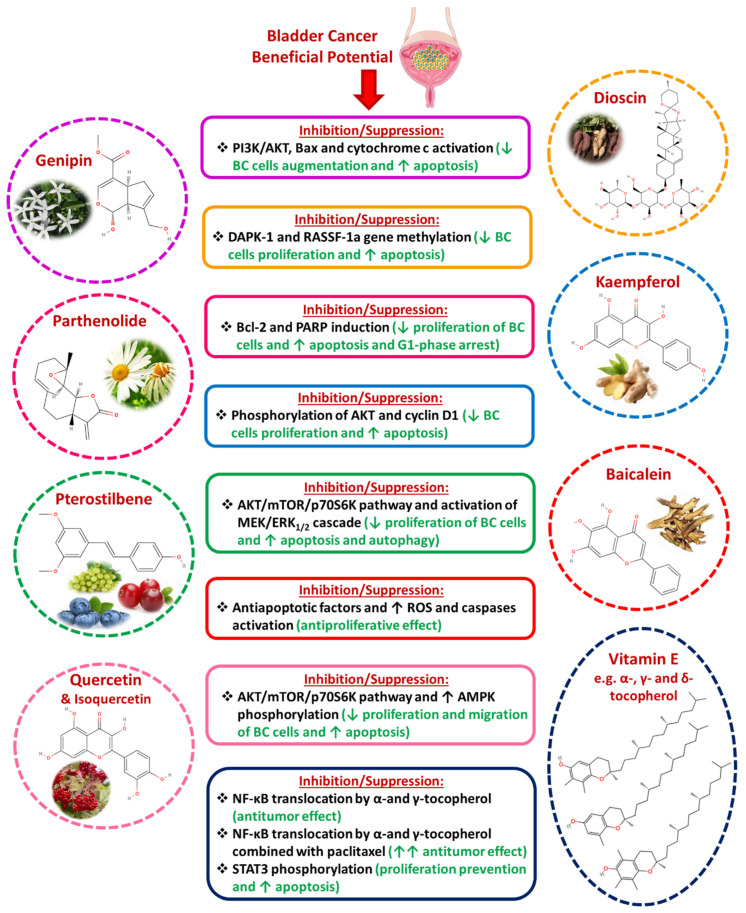
Bladder cancer beneficial potential of several phytochemicals.

**Table 1 diseases-13-00103-t001:** Ongoing trials regarding the use of phytochemicals in BC based on the database clinicaltrials.gov.

Phytochemical	Number of Patients	Inclusion Criteria	Exclusion Criteria	Dose	Duration	Endpoints	Identification Number
Genistein	supplement (N = 44) or placebo (N = 44)	>18 years or older Superficial bladder cancer Programmed BCG intravesical therapy	Pregnancy, muscle-invasive bladder cancer HIV-immunocompromised Concurrent immune or chemotherapy Concurrent active second cancer	30 mg x3	10 weeks	Reducing toxicity and enhancing the efficacy of intravesical therapy, urinary symptoms, recurrence	NCT01489813
Curcumin	Single group	Ureteral stent Patient reports pain, spasms, or urgency symptoms after stent placement, English knowledge Willingness to participate in a follow-up visit Willingness to provide mandatory 24-h urine collection Able to swallow supplements History of cancer or active cancer Registration ≥ 7 days after placement of a new stent or ≥3 days after a stent exchange; Willingness to refrain from grapefruit juice for 7 days before and for 7 days during the study	Warfarin at registration; Active cholecystitis The following drugs: epidermal growth factor receptor inhibitor, topoisomerase 1 inhibitor, buspirone, benzodiazepines, zolpidem, calcium channel blockers; digoxin or quinidine; codeine or fentanyl; phenytoin, propranolol, rifampin, or theophylline; History of alcohol abuse	Varies (dose-response study)	7 days, 1 month follow-up	Adverse events Maximum tolerated dose of curcumin in combination with piperine Optimal biologically active dose for curcumin in combination with piperine extract quality of life Change in prostaglandin E2 concentrations	NCT02598726

The clinical relevance of the use of phytochemicals is also evident through their research testing in other types of cancer, as presented in Appendix A. Although data are limited, it is shown that phytochemicals may alleviate treatment’s side effects [173,174,175,176,177,178,179,180,181,182,183,184,185,186,187,188,189,190,191,192,193].

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
