# Peer review of "Phytochemicals as Chemo-Preventive and Therapeutic Agents Against Bladder Cancer: A Comprehensive Review"

_diseases, 2025, doi:10.3390/diseases13040103_

Round 1
Reviewer 1 Report
Comments and Suggestions for Authors
Comments and Suggestions for Authors
The review article is well written, however for the most part the authors have simply repeated the conclusions of the studies, i.e., critical assessment of the literature cited is lacking. There are several suggestions for the authors before publication.
- The discussion talks about the importance of bioavailability and dose of phytochemicals but does not explore thoroughly how these factors affect their therapeutic potential.
- More discussion on the pharmacokinetics of these compounds would make the review more thorough.
- This review discusses multiple molecular pathways that are implicated in bladder cancer but does not clarify the clinical relevance of these molecular pathways, more discussion of the molecular pathways and its clinical relevance to bladder cancer would strengthen this review.
Author Response
The review article is well written, however for the most part the authors have simply repeated the conclusions of the studies, i.e., critical assessment of the literature cited is lacking. There are several suggestions for the authors before publication.
Thank you for all your recommendations, all the changes are figured in yellow in the paper
- The discussion talks about the importance of bioavailability and dose of phytochemicals but does not explore thoroughly how these factors affect their therapeutic potential.
Thank you for the comment, information concerning the bioavailability and doses was added for each phytochemical mentioning also the strategies used to increase their bioavailability lines( 320-335), (387-402), (434-440), (464-471), (516-521), (567-570), (685-689)
- More discussion on the pharmacokinetics of these compounds would make the review more thorough.
Information regarding the pharmacokinetics was added for each phytochemical lines ( 320-335), (387-402), (434-440), (464-471), (516-521), (567-570), (685-689)
- This review discusses multiple molecular pathways that are implicated in bladder cancer but does not clarify the clinical relevance of these molecular pathways, more discussion of the molecular pathways and its clinical relevance to bladder cancer would strengthen this review.
Thank you for the comment, clinical relevance to bladder cancer was added to each molecular pathway Lines: (94-100), (114-121), (130-137), (152-158), (171-177), (190-194), (215-218), (233-235), (245-249), (255-258), (259-262), (268-272), (276-279), (292-294), (307-311)
Reviewer 2 Report
Comments and Suggestions for Authors
This manuscript is well-written, with high-quality figures. However, several improvements are required to strengthen its impact and clarity.
1. Expand the introduction section by including the comparison between current chemotherapeutic agents and phytochemicals.
2. Restructure Figures 3 and 4 by simplifying the words as they contain excessive words currently.
3. There are two sections labeled “3.4”. Please check and correct this issue.
4. Although quite a lot of phytochemicals are discussed, the novelty and clinical significance of this manuscript remain unclear to me. Currently, there are limited ongoing and completed clinical trials, which means that the clinical translational potential of phytochemicals in BC treatment is restricted.
5. The paper lists 14 natural products. Are these the only natural products for BC treatment, or were these selected based on specific criteria? Please justify the selection.
6. The treatment failure of current chemotherapy may also be linked to multidrug resistance (MDR). Are any phytochemicals associated with overcoming MDR? If so, please include a section discussing MDR.
Author Response
This manuscript is well-written, with high-quality figures. However, several improvements are required to strengthen its impact and clarity.
Thank you for all your recommendations, all the changes are figured in yellow in the paper
- Expand the introduction section by including the comparison between current chemotherapeutic agents and phytochemicals.
A extended paragraph was added to the introduction emphasizing the differences between both approaches as well as the their possible synergistic action (lines 51-69)
- Restructure Figures 3 and 4 by simplifying the words as they contain excessive words currently.
Thank you for your comments the figures 3 and 4 were restructure
- There are two sections labeled “3.4”. Please check and correct this issue.
done
- Although quite a lot of phytochemicals are discussed, the novelty and clinical significance of this manuscript remain unclear to me. Currently, there are limited ongoing and completed clinical trials, which means that the clinical translational potential of phytochemicals in BC treatment is restricted.
Thank you for your comments. Recent clinical trials using the mentioned phytochemicals are presented in supplementary table 1 emphasizing the clinical potential of the mentioned phytochemicals. Also more information about bioavailability, pharmacokinetics, strategies used for increasing the bioavailability, toxicity, drug interaction and MDR of the phytochemicals were added to the paper.
- The paper lists 14 natural products. Are these the only natural products for BC treatment, or were these selected based on specific criteria? Please justify the selection.
Thank you for your remark. The selected phytochemicals have evidence-based data supporting their potential use in supporting patients with bladder cancer, and there is data on their action on the molecular pathways involved in bladder cancer progression
- The treatment failure of current chemotherapy may also be linked to multidrug resistance (MDR). Are any phytochemicals associated with overcoming MDR? If so, please include a section discussing MDR.
Thank you comment, information concerning MDR are depicted in the discussion line 691 - 704
Reviewer 3 Report
Comments and Suggestions for Authors While this is a scientific text, some pathways and proteins (e.g., "Trop2," "Kelch-like (erythroid cell-derived protein with cap ‘n’ collar (CNC) homology (ECH))-associated protein 1") are listed without sufficient background for a broader scientific audience. A brief explanation of their relevance would be helpful. While bioavailability is mentioned in some cases (e.g., resveratrol), the discussion does not extend to potential clinical solutions or formulations (e.g., nanoformulations, prodrug strategies) that could improve therapeutic effectiveness. The focus is heavily on molecular mechanisms, with little discussion of in vivo studies, clinical trials, or real-world applications. Adding information on dosage, patient outcomes, and limitations in clinical settings would strengthen the analysis. The potential toxicity, adverse effects, and interactions with other drugs or treatments are not adequately discussed, which is critical for evaluating therapeutic feasibility.
While preclinical findings are detailed, little emphasis is placed on how these findings translate to actual patient treatment. Highlighting clinical trials, ongoing research, or FDA-approved formulations would add more practical value.
Author Response
Thank you for all your recommendations, all the changes are figured in yellow in the paper
While this is a scientific text, some pathways and proteins (e.g., "Trop2," "Kelch-like (erythroid cell-derived protein with cap ‘n’ collar (CNC) homology (ECH))-associated protein 1") are listed without sufficient background for a broader scientific audience. A brief explanation of their relevance would be helpful.
Brief comment about the role of these molecules in cancer was added (line 367-347)
While bioavailability is mentioned in some cases (e.g., resveratrol), the discussion does not extend to potential clinical solutions or formulations (e.g., nanoformulations, prodrug strategies) that could improve therapeutic effectiveness. The focus is heavily on molecular mechanisms, with little discussion of in vivo studies, clinical trials, or real-world applications.
Thank you for the comment, information concerning the bioavailability and doses was added for each phytochemical mentioning also the strategies used to increase their bioavailability. lines ( 320-335), (387-402), (434-440), (464-471), (516-521), (567-570), (685-689). Clinical trials in bladder cancer was mentioned in the previous version. Examples of clinical trials in different type of cancers were presented in supplementary table 1 in order to deepen into clinical applications.
Adding information on dosage, patient outcomes, and limitations in clinical settings would strengthen the analysis. The potential toxicity, adverse effects, and interactions with other drugs or treatments are not adequately discussed, which is critical for evaluating therapeutic feasibility. While preclinical findings are detailed, little emphasis is placed on how these findings translate to actual patient treatment. Highlighting clinical trials, ongoing research, or FDA-approved formulations would add more practical value.
Thank you for your comments potential toxicity, adverse effects, and interactions with other drugs or treatments are mentioned for each mentioned phytochemical (Lines: 380-382, 423-430, 451-455, 510-512, 560-562, and discussed in discussion (lines: 685-690, 708-711, 723-727).
Recent clinical trials using the mentioned phytochemicals are presented in supplementary Table 1, emphasizing their clinical potential. The paper also includes more information about bioavailability, pharmacokinetics, strategies used for increasing bioavailability, toxicity, drug interaction, and MDR of the phytochemicals.
Round 2
Reviewer 2 Report
Comments and Suggestions for Authors
The authors have addressed my comments point-by-point accordingly. I would recommend the acceptance of this manuscript. Thank you.
Author Response
Thank you for the comments.
Reviewer 3 Report
Comments and Suggestions for Authors
I think this paper has include the previous recomandations. My only question is concerning the resveratrol doses.
I want to send you the fragment from the text which I think must be explained : While resveratrol is generally considered safe and generally well tolerated at lower doses (up to 5 gr), higher doses (100 mg or higher) especially over extended periods, may lead to adverse effects, predominantly affecting the gastrointestinal system [135].
Author Response
I think this paper has include the previous recommendations. My only question is concerning the resveratrol doses.
I want to send you the fragment from the text which I think must be explained : While resveratrol is generally considered safe and generally well tolerated at lower doses (up to 5 gr), higher doses (100 mg or higher) especially over extended periods, may lead to adverse effects, predominantly affecting the gastrointestinal system [135].
Thank you for the comments, this paragraph was changed as below: (depicted in green)
According to clinical studies, resveratrol is generally considered safe and well tolerated as a single dose up to 450 mg /day for a 60kg person without adverse effects. However, 1000 mg daily or higher doses were reported to inhibit cytochrome P450 enzymes, leading to interactions with many drugs. Higher doses were shown to elevate some biomarkers of CDV risk (ox-LDL, sE-Selectin 1, sICAM-1, sVCAM, tPAI-1) and affect the gastrointestinal system. These adverse effects could vary according to individual gut microbiota, health status, gender, lifestyle and the form of administration (with or without food, caplet, tablet, powder etc) [135].